# SOBOLEV ACCELERATION FOR NEURAL NETWORKS

## ABSTRACT

Sobolev training for neural networks, a technique that integrates target derivatives into the training process, has demonstrated significantly faster convergence towards lower test errors when compared to conventional loss functions. However, to date, the effect of this training has not been understood comprehensively. This paper presents analytical evidence that Sobolev training accelerates the convergence of rectified linear unit (ReLU)-networks in the student–teacher framework. The analysis builds upon the analytical formula for the population gradients of ReLU networks with centered spherical Gaussian input. Further, numerical examples were considered to show that the results may be extended to multi-layered neural networks with various activation functions and architectures. Finally, we propose the use of Chebyshev spectral differentiation as a solution to approximate target derivatives and address prior limitations on using approximated derivatives. Overall, this study contributes to a deeper understanding of the dynamics of ReLU networks in the student–teacher setting and highlights the convergence acceleration achieved through Sobolev training, known as Sobolev acceleration.

## 1 INTRODUCTION

In recent years, deep learning has witnessed tremendous growth and development, achieving remarkable success across various scientific and engineering domains. This success can be attributed to the development of novel deep neural network architectures such as U-Net Ronneberger et al. (2015), ResNet He et al. (2016), AlexNet Krizhevsky et al. (2017), RNN encoder–decoder Cho et al. (2014), and Transformer Vaswani et al. (2017) and powerful optimization techniques including Adam Kingma & Ba (2014); Ruder (2016) and RMSprop Riedmiller & Braun (1993). These developments have resulted in significant advancements in computer vision Shorten & Khoshgoftaar (2019); Voulodimos et al. (2018) and natural language processing Young et al. (2018); Otter et al. (2020). More recently, reflecting physics in training, deep learning has gained significant attention in the field of applied and computational mathematics, particularly for scientific computing. This development has opened up new avenues for the modeling and simulation of complex physical systems, such as fluid dynamics, materials science, and quantum mechanics, among others Karniadakis et al. (2021).

However, despite these impressive accomplishments, our understanding of the training dynamics of neural networks remains limited. This knowledge gap represents a significant challenge in the field of deep learning and motivates continued research aimed at developing a deeper understanding of the inner workings of these powerful models. As an initial effort to analyze the networks, researchers have focused on their expressive power. Following the seminal work by Cybenko (1989), many studies have improved the density argument for the neural networks in Cybenko (1989); for instance, Hornik et al. (1989) generalized the result to multilayer neural networks and Li (1996) proved the density argument in the Sobolev spaces. In contrast to the concrete knowledge of the density argument, when the gradient descent is considered, the training dynamics of neural networks are only partially discovered owing to their nonconvex nature.

To analytically study the training dynamics of shallow neural networks with rectified linear unit (ReLU) activation under gradient descent, a line of research Tian (2017); Li & Yuan (2017); Zhang et al. (2019) adopts a student–teacher framework, assuming the presence of a ground truth teacher network with the same architecture as that of the student network. Another line of research focuses on overparameterization, including Du et al. (2018); Chizat & Bach (2018); Arora et al. (2019); Allen-Zhu et al. (2019); Zou et al. (2020). Notably, Jacot et al. (2018) introduced the notion that

the training of a neural network can be characterized by a constant kernel, referred to as the neural tangent kernel (NTK), in the infinite width limit. Furthermore, Wang et al. (2022) extended this concept to physics-informed neural networks (PINNs). They derived the NTK for these networks and demonstrated its convergence to a constant kernel.

The authors of Czarnecki et al. (2017) proposed to minimize the Sobolev norms of an error function rather than the $L^2$-norm, and named this training process as Sobolev training. They proved that Sobolev training significantly reduced the sample complexity of training and demonstrated its ability to achieve considerably lower test error compared to the conventional $L^2$-loss function. The impact of Sobolev training has extended across various fields, prompting extensive research. For instance, for PINNs, Son et al. (2023) introduced multiple loss functions tailored to Sobolev training, enhancing the training process. In another application, Vlassis & Sun (2021) harnessed Sobolev training to refine smoothed elastoplasticity models. Kissel & Diepold (2020) proposed to leverage approximated derivatives when the target derivatives are unavailable. The potential of Sobolev training was further exemplified by Cocola & Hand (2020), who demonstrated the global convergence of this approach for overparameterized networks. More recently, Yu et al. (2023) showcased how Sobolev loss functions could effectively manage the spectral bias of neural networks.

This study aimed to establish a theoretical foundation for understanding the accelerated convergence achieved by Sobolev training in comparison to conventional $L^2$ training. While this acceleration phenomenon has been observed in various studies Son et al. (2023); Lu et al. (2022), the existing analytical tools, particularly those related to derivative losses Cocola & Hand (2020); Yu et al. (2023); Wang et al. (2022), are unable to explain the effect comprehensively. In this study, we adopted a student–teacher framework for ReLU networks, building upon the approach introduced in Tian (2017). We established and proved the acceleration effect within the context of gradient flow, specifically for the population loss function defined over spherical Gaussian input data. Further, we proposed the use of Chebyshev spectral differentiation to adopt Sobolev training even when the target derivative was unavailable. Consequently, we empirically showed that the proposed method overcame the limitations of the existing method, that is, the finite difference scheme. Furthermore, we empirically validated the acceleration effect across a range of generalized scenarios, encompassing empirical loss minimization under non-Gaussian inputs, multilayered networks, and diverse activation functions and optimizers.

## 1.1 CONTRIBUTIONS

- We presented a proof of Sobolev acceleration, covering both $H^1$ and $H^2$ norms, for a specific class of ReLU-activated networks in the student–teacher framework. We achieved this by deriving analytical formulas for the population gradient flow of the $L^2$, $H^1$, and $H^2$ norms.

- We illustrated our analysis through numerical examples, thereby demonstrating its generalization to a practical scenario. In particular, empirical risk minimization using stochastic gradient descent (SGD) was demonstrated for various learning rates and batch sizes.

- We empirically demonstrated the Sobolev acceleration as a general phenomenon in training neural networks, considering various activation functions and architectures including the Fourier feature networks Tancik et al. (2020), and SIREN Sitzmann et al. (2020).

- We proposed to leverage the Chebyshev spectral differentiation to approximate target function derivatives, particularly when the target derivative was unavailable. Our experimental results demonstrated that the proposed method surpassed existing finite difference schemes.

- We also applied Sobolev training for training the denoising autoencoder and demonstrated both convergence acceleration and improved generalization.

## 2 THEORETICAL RESULTS IN SOBOLEV ACCELERATION

### 2.1 SOBOLEV TRAINING

Regression problems aim to minimize the error between the hypothesis function(in our case, a neural network) and the target function. In other words, a minimization problem is solved

$$\underset{\theta}{\text{minimize}} \, \mathbb{E}_{x \sim \mathcal{P}} \left[ (u_\theta(x) - f(x))^2 \right] \approx \underset{\theta}{\text{minimize}} \, \frac{1}{N} \sum_{i=1}^{N} |u_\theta(x_j) - f(x_j)|^2,$$

where $u_\theta$ is a hypothesis, $f$ is a target function, and $\mathcal{P}$ is a data distribution. However, Sobolev training aims to minimize both the expected squared difference and the expected squared difference of derivatives

$$\underset{\theta}{\text{minimize}} \, \mathbb{E}_{x \sim \mathcal{P}} \left[ (u_\theta(x) - f(x))^2 + |\nabla_x u_\theta(x) - \nabla_x f(x)|^2 \right]$$

$$\approx \underset{\theta}{\text{minimize}} \, \frac{1}{N} \sum_{i=1}^{N} \left[ |u_\theta(x_j) - f(x_j)|^2 + |\nabla_x u_\theta(x_j) - \nabla_x f(x_j)|^2 \right].$$

In Czarnecki et al. (2017), the authors provided evidence that Sobolev training reduces the sample complexity of training and achieved considerably higher accuracy and stronger generalization. Later, Lu et al. (2022) demonstrated implicit Sobolev acceleration, and Son et al. (2023) showed that Sobolev training expedited the training of neural networks for regression and PINNs. In this section, we theoretically confirm this acceleration effect of Sobolev training.

We assume the presence of a teacher network to facilitate the computation of the derivatives of the target. Without this assumption, proving the acceleration of Sobolev training becomes challenging owing to the absence of relational information between the target and its derivative. For example, Cocola & Hand (2020) proved the convergence of Sobolev training in the NTK regime. However, as the labels for the target and its derivative were defined as separate vectors, this approach could not provide insights into the relationship between the two components in the Sobolev loss function, which hindered further derivation of the acceleration results. In contrast, our analysis, while constrained to neural networks with simple architectures, offer a concrete understanding of Sobolev acceleration.

## 2.2 $\mathcal{H}^1$ LOSS WITH RELU ACTIVATION

Let $N$ be the number of samples and $d$ be the input dimension. We assume that the data follows the $d-$dimensional centered spherical Gaussian distribution $X \in \mathbb{R}^{N \times d} \sim N(0, I_{d \times d})$. In this setting, we prove the Sobolev acceleration effect for ReLU-type neural networks in the student–teacher setting, where a student parameter learns a teacher parameter $w^*$, which defines a target function. We compare the dynamics of the error function $V(w) = \|w - w^*\|^2$ for the loss functions defined by different Sobolev norms, $L^2$, $H^1$, and $H^2$ (only in Section 2.3). We compute the analytical formulas of the dynamics under the gradient flow $\dot{w} = -\nabla_w \mathbb{E}_{X \sim N(0,I)}(J(w; X))$ of the population loss function $\mathbb{E}_{X \sim N(0,I)}(J(w; X))$, where $J$ is a loss function. The error dynamics is expressed as follows:

$$\dot{V}(w) = -(w - w^*)^T \nabla_w \mathbb{E}_{X \sim N(0,I)}(J(w; X)).$$

In this section, we prove that the convergence $V \to 0$ is accelerated by the Sobolev loss functions.

We begin by mentioning the convergence result for a single ReLU node.

**Theorem 1** (Theorem 5 in Tian (2017)). *Let $g(x; w) = \sigma(w^T x)$ be a neural network with a single ReLU node, where $w, x \in \mathbb{R}^d$, and $\sigma(x) = \max(0, x)$. We define the population loss function as*

$$\mathcal{L}(w) = \mathbb{E}_X \left( \frac{1}{2N} \sum_{j=1}^{N} (g(x_j; w) - g(x_j; w^*))^2 \right), \tag{1}$$

*for a teacher parameter $w^*$ and consider the gradient flow $\dot{w} = -\nabla_w \mathcal{L}(w)$. If $w^0 \in \{w : \|w - w^*\| < \|w^*\|\}$, then $\frac{dV}{dt} = -(w - w^*)^T \nabla_w \mathcal{L} < 0$ and $w^t \to w^*$ as $t \to \infty$.*

This theorem states that for a neural network with a single ReLU node, global convergence can be achieved depending on the initial parameter $w^0$. In the next theorem, which is our first result, we prove that using the $\mathcal{H}^1$ loss function, the convergence of $V$ can be accelerated to 0.

**Theorem 2.** *Let $g(x; w) = \sigma(w^T x)$ be a neural network with a single ReLU node, where $w, x \in \mathbb{R}^d$, and $\sigma(x) = \max(0, x)$. We define the population loss function in $H^1$ space as*

$$\mathcal{H}(w) = \mathbb{E}_X \left( \frac{1}{2N} \sum_{j=1}^N (g(x_j; w) - g(x_j; w^*))^2 + \|\nabla_x g(x_j; w) - \nabla_x g(x_j; w^*)\|^2 \right),$$

*for a teacher parameter $w^*$ and consider the gradient flow $\dot{w} = -\nabla_w \mathcal{H}(w) =: -\nabla_w(\mathcal{L} + \mathcal{J})$. If $w^0 \in \{w : \|w - w^*\| < \|w^*\|\}$. Then,*

$$\frac{dV}{dt} = -(w - w^*)^T \nabla_w \mathcal{H} < -(w - w^*)^T \nabla_w \mathcal{L} < 0,$$

*where $\mathcal{L}$ is given in 1, and hence, the convergence $w \to w^*$ is accelerated.*

*Proof.* We provide a sketch of the proof, with the complete derivation presented in the Appendix. By definition, $\nabla_w \mathcal{H} = \nabla_w \mathcal{L} + \nabla_w \mathbb{E}(\frac{1}{2}\|\nabla_x g(X; w) - \nabla_x g(X; w^*)\|^2)$. We prove the theorem by computing an analytical formula of the gradient of $H^1$ seminorm term. Note that $\nabla_x g(x; w) = \sigma'(w^T x)w = \mathbb{1}_{w^T x > 0} w$.

$$\nabla_w \mathcal{J} = \frac{(\pi - \theta)}{2\pi}(w - w^*) + \frac{\theta}{2\pi} w,$$

, where $\theta$ denotes the angle between $w$ and $w^*$. Therefore,

$$\frac{dV}{dt} = -(w - w^*)^T(\nabla_w(\mathcal{L} + \mathcal{J})) =: -\begin{pmatrix} \|w^*\| \\ \|w\| \end{pmatrix}^T (M_1 + M_2) \begin{pmatrix} \|w^*\| \\ \|w\| \end{pmatrix}.$$

For $\theta \in (0, \pi/2]$, both $M_1, M_2$ are positive definite, and hence, the conclusion follows. $\square$

## 2.3 $\mathcal{H}^2$ LOSS WITH RELU$^2$ ACTIVATION

We now demonstrate the same effect for higher-order derivatives. As the ReLU function is now twice weakly differentiable, we considered a neural network with a single ReLU-square node, which has been widely considered in the literature Yu et al. (2018); Cai & Xu (2019),

$$g(x) = \left( \sigma(w^T x) \right)^2,$$

where $w, x \in \mathbb{R}^d$. We show the global convergence of the neural network with one ReLU$^2$ node in $L^2$ and the acceleration of the convergence in $H^1$, $H^2$ spaces.

**Theorem 3.** *Let $g(x; w) = \left( \sigma(w^T x) \right)^2$ be a neural network with a single ReLU$^2$ node, where $w, x \in \mathbb{R}^d$, and $\sigma(x) = \max(0, x)$. We define the population loss function in $H^2$ space by*

$$\mathcal{I}(w) = \mathbb{E}_X \left( \frac{1}{2N} \sum_{j=1}^N (g(x_j; w) - g(x_j; w^*))^2 + \|\nabla_x g(x_j; w) - \nabla_x g(x_j; w^*)\|^2 \right.$$

$$\left. + \|\nabla_x^2 g(x_j; w) - \nabla_x^2 g(x_j; w^*)\|^2 \right),$$

$$=: \mathcal{I}_1(w) + \mathcal{I}_2(w) + \mathcal{I}_3(w),$$

*for a teacher parameter $w^*$ and consider the gradient flow $\dot{w} = -\nabla_w \mathcal{I}(w)$. If $w^0 \in \{w : \|w - w^*\| < \|w^*\|\}$. Then,*

$$-(w - w^*)^T \nabla_w \mathcal{I}_j(w) < 0, \text{ for } j = 1, 2, 3,$$

*and hence, the convergence of $V = \|w - w^*\|^2$ is accelerated under the gradient flow that minimizes the higher order Sobolev loss functions.*

*Proof.* The strategy is nearly identical to that in the proof of Theorem 2. We computed the analytical formula for the population gradients $\nabla_w \mathcal{I}_j$ for $j = 1, 2$, and, $3$, and compared the gradient flows of $V = \|w - w^*\|^2$. The complete proof is presented in the Appendix. $\square$

# 3 SOBOLEV TRAINING WITH CHEBYSHEV SPECTRAL DIFFERENTIATION

One major obstacle to applying Sobolev training is that it requires additional derivative information. However, recent studies have reported that Sobolev training works well even when the derivative information is unavailable by relying on approximated derivatives obtained through numerical techniques. For example, Kissel & Diepold (2020) applied the finite difference scheme, and Yu et al. (2023) utilized spectral differentiation to approximate derivatives in the context of Sobolev training for autoencoders. However, the finite difference scheme is vulnerable to the combination of $L^2$ and $H^1$ seminorm loss functions, such that one of the two often dominates the other and hinders the training. In addition, the spectral differentiation used in Yu et al. (2023) is built upon the periodicity assumption of the solution. To overcome these limitations, we proposed to leverage the Chebyshev spectral differentiation Trefethen (2000), which is among the most successful numerical differentiation methods. The Chebyshev spectral differentiation can be implemented by multiplying the differentiation matrix $D_{Chev}$ presented in Trefethen (2000).

The optimization problem for Sobolev training is

$$\underset{\theta}{\text{minimize}} \frac{1}{N} \sum_{i=1}^{N} \left[ |u_\theta(x_j) - f(x_j)|^2 + |\nabla_x u_\theta(x_j) - \nabla_x f(x_j)|^2 \right]$$

This becomes

$$\underset{\theta}{\text{minimize}} \frac{1}{N} \left( \sum_{i=1}^{N} \left[ |u_\theta(x_j) - f(x_j)|^2 \right] + \|vec(\nabla_x u_\theta(x_j)) - D_{Chev} vec(f(x_j))\|_2^2 \right),$$

for the Chebyshev nodes $\{x_j\}_{j=1}^N$. The Chebyshev spectral differentiation can also be implemented using the fast Fourier transform (FFT), as described in Trefethen (2000).

# 4 EXPERIMENTS

## 4.1 ANALYTICAL FORMULAS FOR THE POPULATION GRADIENTS

We verified the analytical formulas for the population gradients presented in Section 2. We randomly sampled $w^*$ from the standard normal distribution and added a uniform random vector $e$ s.t. $\|e\| \leq \|w^*\|$ to $w^*$ to obtain $w = w^* + e$ and $\|w - w^*\| \leq \|w^*\|$. We employed two neural networks

$$g_i(x; w) = (\sigma(w^T x))^i, \text{ and } g_i(x; w^*) = (\sigma(w^{*T} x))^i,$$

for $i = 1, 2$ with parameters $w$, and $w^*$, respectively. Subsequently, the error between the analytical formula (e.g., $\nabla_w \mathcal{J}, \nabla_w \mathcal{I}_j$) and the Monte-Carlo approximation of the population loss under spherical Gaussian distribution was computed by varying the input dimensions and the number of samples. Figure 2 shows the log–log plots for mean square errors between the analytical formulas and the Monte-Carlo approximations. For example, we computed $\frac{1}{2N} \sum_{i=1}^{N} \nabla_w \|\nabla_x g(x_j; w) - \nabla_x g(x_j; w^*)\|_2^2$ using automatic differentiation, and computed its discrepancy to $\nabla_w \mathcal{J}$. In all cases, the error decreased linearly in the log-log scale, as the number of samples increased. Moreover, the errors were sufficiently small even in relatively high dimensions.

## 4.2 EMPIRICAL RISK MINIMIZATION UNDER STOCHASTIC GRADIENT DESCENT

Our theoretical results relied on the gradient flow dynamics of the population loss function, assuming an infinitesimal learning rate. However, in practical settings, we often use stochastic gradient descent(SGD), which incorporates mini-batch gradient descent of the empirical loss function, with relatively large learning rates. In this subsection, we demonstrate through numerical examples that the Sobolev acceleration effect persists in empirical loss minimization with SGD.

We again randomly considered $w^* \in \mathbb{R}^d$ and selected $w$ such that $\|w - w^*\| \leq \|w^*\|$. We used SGD to minimize the empirical loss functions $\frac{1}{2N} \sum_{i=1}^{N} (g(x_j; w) - g(x_j; w^*))^2$ and $\frac{1}{2N} \sum_{i=1}^{N} (g(x_j; w) - g(x_j; w^*))^2 + \|\nabla_x g(x_j; w) - \nabla_x g(x_j; w^*)\|^2$, where $g(x; w) = \sigma(w^T x)$

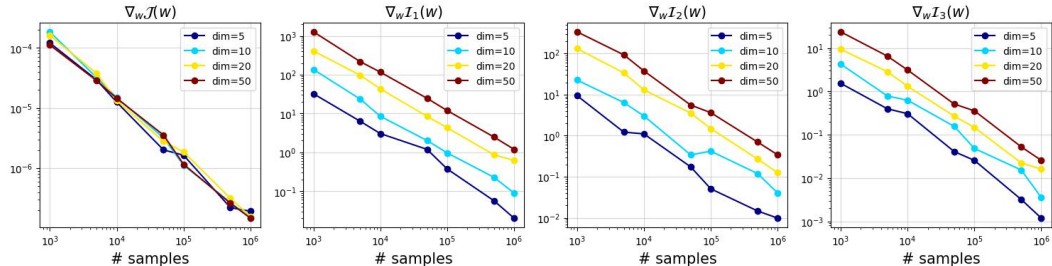

Figure 1: Log–log plots of the mean square errors(MSE) versus the number of samples. MSEs are computed between the analytical formulas $\nabla_w \mathcal{J}, \nabla_w \mathcal{I}_j$ and empirical expected values. Errors tend to decrease as the number of samples increase across all input dimensions.

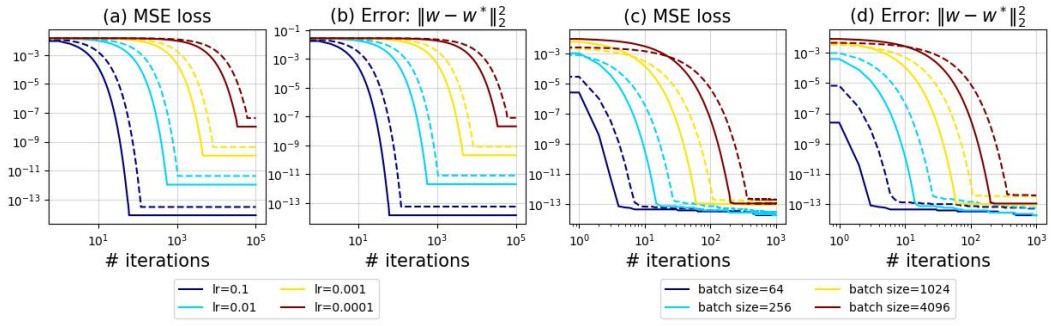

Figure 2: Comparison of convergence of $L^2$ training(dashed lines) and $H^1$ training(solid lines). (a), (b): Losses and errors for different learning rates. (c), (d): Losses and errors for different batch sizes. In all cases, Sobolev training clearly accelerates the convergence of both loss and error. We can also observe that Sobolev training achieves a better local minimum in terms of both loss and error.

and N=10,000. We explored a range of relatively large learning rates: [1e-1, 1e-2, 1e-3, 1e-4]. Figures 1(a) and (b) illustrate the MSE loss values and the errors $\|w - w^*\|$, respectively, during training for various learning rates in log–log scales. Figures 1 (c) and (d) show the MSE loss values and the errors, respectively, for different batch sizes in [64, 256, 1024, 4096]. As evident, Sobolev training expedited convergence and yielded superior local minima in terms of both loss and error.

## 4.3 SOBOLEV ACCELERATION FOR VARIOUS ARCHITECTURES

**Setup.** We first present empirical evidence that the Sobolev acceleration is a general phenomenon occurring across various activation functions and architectural setups and not limited to the student–teacher setting under Gaussian input with a single ReLU or ReLU$^2$ node. In other words, we now consider the following generic ($L^2-$)regression problem:

$$\underset{w}{\text{minimize}} \; \frac{1}{2N} \sum_{j=1}^{N} \left( u(x_j; w) - f(x_j) \right)^2,$$

and Sobolev training for the same problem:

$$\underset{w}{\text{minimize}} \; \frac{1}{2N} \sum_{j=1}^{N} \left( u(x_j; w) - f(x_j) \right)^2 + \| \nabla_x u(x_j; w) - \nabla_x f(x_j) \|_2^2,$$

where $f$ is a target function, for various neural networks $u(x; w)$.

**Various activations** As an initial illustrative example, we compared the Sobolev acceleration for standard fully connected neural networks with different activation functions. The target function

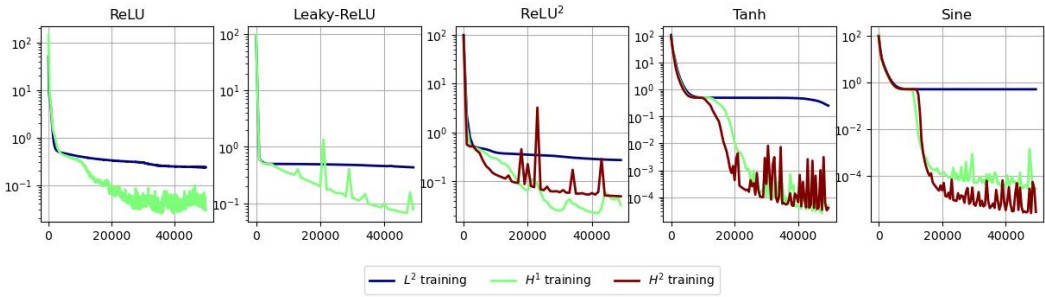

Figure 3: MSE versus training iterations for different activation functions. Each network comprises three hidden layers, each of which has 64 hidden nodes. We train the networks for 50,000 training epochs. $H^1$ training accelerates convergence in all cases, while the $H^2$ loss function enhances training specifically for the sine activation function.

was set as $f(x, y) = \sin(10 * (x + y)) + (x - y)^2 - 1.5x + 2.5y + 1$ on $(x, y) \in [1, 4] \times [-3, 4]$ and a fully connected neural network with 2-64-64-64-1 hidden nodes was employed along with an ADAM(Kingma & Ba (2014)) optimizer. The learning rate was 1e-4 to minimize the loss functions for 50,000 epochs. We selected ReLU, Leaky ReLU, ReLU-squared, Tanh, and Sine functions as the activation functions. The results are presented in Figure 3. For all activation functions, Sobolev training with $H^1$ loss function resulted in a significantly faster convergence than the $L^2$ training. However, the $H^2$ loss function expedited the training only for the sine activation function. Notably, Sobolev training was considerably more powerful when applied with the sine activation function. As reported by Sitzmann et al. (2020), the periodic activation function enables the network to learn the high-frequency features. We believe that the Sobolev loss function intensifies the sine-activated network to be more robust in learning high-frequency features of the target function as discussed in a recent study Yu et al. (2023).

**Sobolev Training for Fourier feature networks and SIRENs** Tancik et al. (2020) proved that standard neural networks prefer to learn low-frequency components than the higher ones. They proposed Fourier feature networks, which were very simple but powerful architectures that explicitly incorporated sinusoidal features before the first hidden layer, to overcome this "spectral bias." Spectral bias is also considered in the Sobolev training literature, as reported by Yu et al. (2023). The loss function with the Sobolev norm can modulate the spectral bias in training neural networks. We demonstrated that Fourier feature networks trained using Sobolev loss functions exhibit significantly greater robustness to spectral bias compared to those trained using the conventional $L^2$ loss function. Another line of research, known as SIREN, was introduced in Sitzmann et al. (2020), which utilized periodic activation functions in conjunction with a principled initialization. SIREN has been proven to robustly capture complex signals and derivatives and we expect Sobolev training to solidify the robustness of SIRENs.

Here, we demonstrated a substantial improvement achieved by combining Sobolev training and the Fourier features and SIREN in solving regression problems on two multi-scale functions. We used the target functions $f_1(x) = \sin(2\pi x) + \sin(20\pi x)$, on $[-1, 1]$, and $f_2(x) = x + \sin(2\pi x^4)$ on $[0, 2]$ which is known to be challenging for neural networks to learn Wang et al. (2021). To build the Fourier feature networks, we built a fully connected network with 64-64-1 hidden nodes upon 64 Fourier features with randomly generated frequencies. To implement SIREN, we used a fully connected network with 1-64-64-64-1 nodes under uniform initialization and sine activation function as in Sitzmann et al. (2020). Figure 4 shows the acceleration of Sobolev training for the Fourier feature networks or SIREN. For the standard MLP, $H^1$ training barely accelerated the convergence in learning $f_1$ whereas the acceleration was observed in the early stage of training for $f_2$. $H^1$ training for the standard MLP yielded an error level similar to that of $L^2$ trained Fourier feature network and SIREN. Moreover, Sobolev training for Fourier features and SIREN converged rapidly to much smaller errors compared to $L^2$-trained standard MLP.

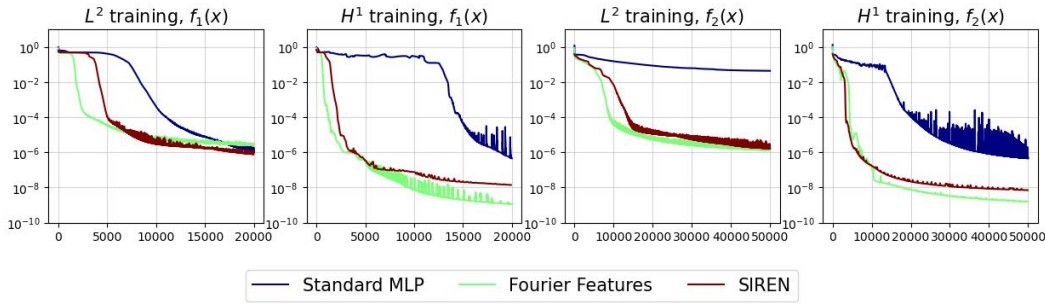

Figure 4: MSE versus training epochs for different architectures. To alleviate the randomness and oscillatory behavior of the errors, we repeatedly trained 100 networks and averaged the errors. Three architectures are compared: the standard MLP, Fourier feature networks, and SIREN.

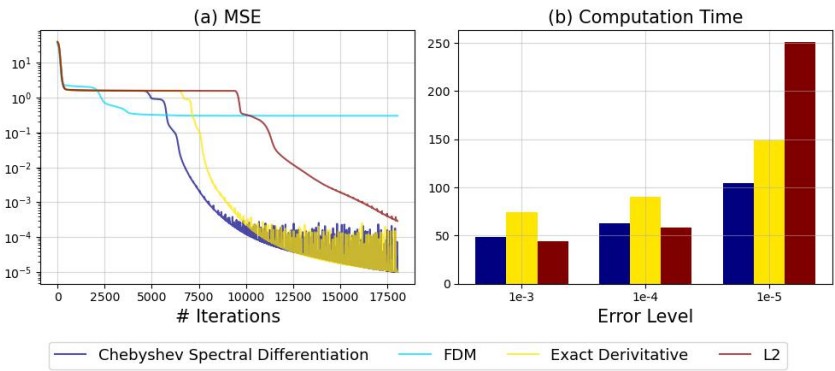

Figure 5: (a): MSEs for different loss functions during training. We compared four loss functions: $L^2$, exact $H^1$, FDM-based $H^1$, and the Chebyshev spectral differentiation based $H^1$. (b): Actual computation times to achieve certain error levels of [1e-3, 1e-4, 1e-5].

### 4.4 SOBOLEV TRAINING WITH APPROXIMATED DERIVATIVES

We now present comparative experiments demonstrating that Sobolev training with approximated derivatives using the proposed Chebyshev spectral differentiation outperformed the finite difference method (FDM) and sometimes exact derivatives. We considered a target function, the Acklev function $f(x, y) = -20 \exp(-0.2\sqrt{0.5(x^2 + y^2)}) - \exp(0.5(\cos(2\pi x) + \cos(2\pi y))) + e + 20$, $(x, y) \in [-2, 2]^2$, from Czarnecki et al. (2017). We trained a neural network of 2-64-64-64-1 nodes with the hyperbolic tangent activation function using the ADAM optimizer with a learning rate of 1e-4. Figure 5 (a) shows the errors during training in log scale for different loss functions: $L^2$, exact $H^1$, $H^1$ based on FDM, and $H^1$ based on the Chebyshev spectral differentiation. The proposed method achieved error levels almost equivalent to those in the exact derivative case, thereby suggesting that we can leverage the benefits of Sobolev training without requiring additional derivative information. Surprisingly, our method exhibited slightly faster convergence than the exact derivative case for this target function. In our experiments, the FDM-based approach converged to an undesired local minimum, specifically a constant solution that resulted in a large $L^2$ loss and zero $H^1$ seminorm loss. The actual computation times required to achieve specific error levels of [1e-3, 1e-4, 1e-5] are presented in Figure 5 (b) for these loss functions. Owing to the efficient computation of tensor multiplication, our method achieved error levels of [1e-3, 1e-4] without significantly increasing computation time compared to the $L^2$ loss function, and reached 1e-5 MSE considerably faster than the $L^2$ loss function.

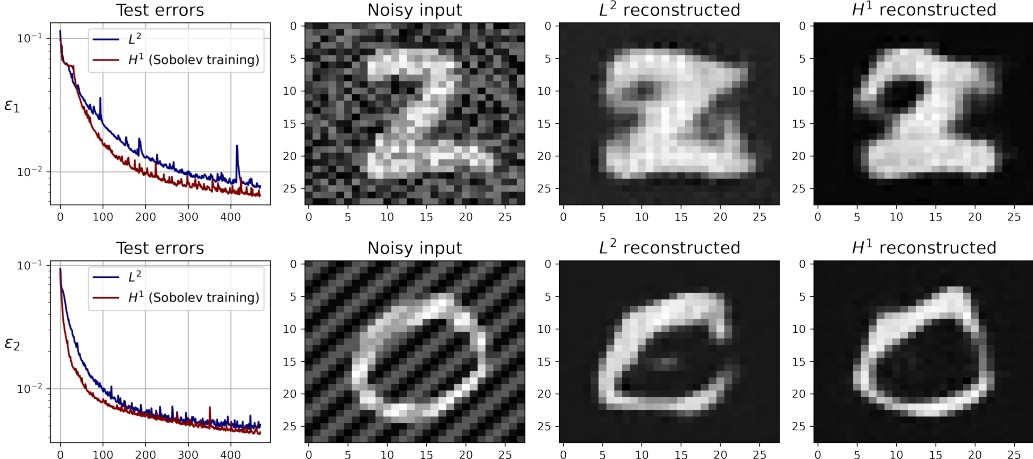

Figure 6: Results of the denoising autoencoders for the $\epsilon_1$ noise case are presented in the top row, and those for the $\epsilon_2$ noise case are shown in the bottom row. The first column illustrates convergence acceleration, the second column displays test inputs with amplified noises, and the third and fourth columns present the corresponding reconstruction results.

## 4.5 SOBOLEV TRAINING FOR THE DENOISING AUTOENCODERS

Autoencoders can be applied to image-denoising tasks by taking noisy input images and training them to output the corresponding original images. This task can be naturally integrated with Sobolev training, as first considered in Yu et al. (2023). We present several numerical experiments that demonstrate the accelerated convergence and improved generalization ability achieved through Sobolev training using the denoising autoencoders equipped with Convolutional Neural Networks (CNNs).

We utilize a simple autoencoder comprising an encoder and a decoder, each consisting of three convolution layers with LeakyReLU activations. The Adam optimizer with a learning rate of 5e-3 is employed. The input image is contaminated with two types of noise: $\epsilon_1 \sim N(0, \sigma_1^2)$, and $\epsilon_2 = \sigma_2 \sin(2\pi\eta(x+y))$, where $x, y \in [0, 1]$, $\epsilon_1$ follows a normal distribution, and $\epsilon_2$ is deterministic noise with a specific amplitude and frequency.

The first column of Figure 6 illustrates the convergence acceleration achieved through Sobolev training in both noise settings. The second column shows the noisy inputs generated from the MNIST dataset (LeCun et al., 2010). During training, the autoencoders are exposed to noisy inputs generated by adding $\epsilon_1 \sim N(0, 1/4)$ and $\epsilon_2 = 0.3 \sin(2\pi(x + y))$. Subsequently, the trained autoencoders are tested with significantly amplified noise levels: $\epsilon_1 \sim N(0, 1)$ and $\epsilon_2 = 0.3 \sin(20\pi(x + y))$. This testing phase aims to assess the improved generalization performance of Sobolev training. In Figure 6, the second column displays the test noisy images, while the third and fourth columns showcase the test reconstruction results of $L^2$ and $H^1$ trained autoencoders, respectively. These results highlight the enhanced generalization ability achieved through Sobolev training.

## 5 CONCLUSION

Sobolev acceleration is a convergence acceleration phenomenon of training neural networks that has been empirically observed in previous studies. Although restricted to a relatively simple architecture, this paper presents the first rigorous theoretical evidence of Sobolev acceleration by considering the gradient flow dynamics of the student–teacher setting. Not limiting ourselves to theoretical findings, we presented several empirical observations that implied that Sobolev acceleration is a general phenomenon occurring in various architectures, including the Fourier features and SIREN. Additionally, we proposed to leverage the Chebyshev spectral differentiation, which can achieve spectral accuracy, to approximate the target derivative for use in Sobolev training. We demonstrated that the proposed method significantly improved the error of various regression problems and overcame the

limitations of the finite difference scheme. Finally, we provide more practical experiments with the MNIST dataset demonstrating both the acceleration and improved generalization of Sobolev training. As a concluding remark, we intend to delve deeper into the analysis of the gradient flow of Sobolev loss functions for neural networks with various architectures in our future work.

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

## A APPENDIX

### A.1 PROOF OF THEOREMS

**Theorem A.1** (Theorem 2). *Let $g(x; w) = \sigma(w^T x)$ be a neural network with a single ReLU node, where $w, x \in \mathbb{R}^d$, and $\sigma(x) = \max(0, x)$. We define the population loss function in $H^1$ space as*

$$\mathcal{H}(w) = \mathbb{E}_X \left( \frac{1}{2N} \sum_{j=1}^N (g(x_j; w) - g(x_j; w^*))^2 + \|\nabla_x g(x_j; w) - \nabla_x g(x_j; w^*)\|^2 \right),$$

*for a teacher parameter $w^*$ and consider the gradient flow $\dot{w} = -\nabla_w \mathcal{H}(w) =: -\nabla_w(\mathcal{L} + \mathcal{J})$. If $w^0 \in \{w : \|w - w^*\| < \|w^*\|\}$. Then,*

$$\frac{dV}{dt} = -(w - w^*)^T \nabla_w \mathcal{H} \leq -(w - w^*)^T \nabla_w \mathcal{L} < 0,$$

*where $\mathcal{L}$ is given in 1, and hence, the convergence $w \to w^*$ is accelerated.*

*Proof.* By definition, $\nabla_w \mathcal{H} = \nabla_w \mathcal{L} + \nabla_w \mathbb{E}(\frac{1}{2} \|\nabla_x g(X; w) - \nabla_x g(X; w^*)\|^2)$. We prove the theorem by computing an analytical formula of the gradient of $H^1$ seminorm term. Note that $\nabla_x g(x; w) = \sigma'(w^T x) w = \mathbb{1}_{w^T x > 0} w$.

$$\nabla_w \mathcal{J} := \nabla_w \mathbb{E} \left( \frac{1}{2N} \|\nabla_x g(X; w) - \nabla_x g(X; w^*)\|^2 \right)$$

$$= \nabla_w \mathbb{E} \left( \frac{1}{2N} \sum_{j=1}^N \|\mathbb{1}_{w^T x_j > 0} w - \mathbb{1}_{w^{*T} x_j > 0} w^*\|^2 \right)$$

$$= \mathbb{E} \left( \frac{1}{N} \sum_{j=1}^N (\mathbb{1}_{w^T x_j > 0} w - \mathbb{1}_{w^T x_j > 0} \mathbb{1}_{w^{*T} x_j > 0} w^*) \right)$$

$$= \frac{1}{N} \sum_{j=1}^N \left( \mathbb{P}(w^T x_j > 0) w - \mathbb{P}(w^T x_j > 0 \wedge w^{*T} x_j > 0) w^* \right)$$

$$= \frac{(\pi - \theta)}{2\pi} (w - w^*) + \frac{\theta}{2\pi} w,$$

where $\theta$ denotes the angle between $w$, and $w^*$. Therefore,

$$\frac{dV}{dt} = -(w - w^*)^T (\nabla_w(\mathcal{L} + \mathcal{J}))$$

$$= -(w - w^*)^T \nabla_w \mathcal{L} - (w - w^*)^T \left( \frac{(\pi - \theta)}{2\pi} (w - w^*) + \frac{\theta}{2\pi} w \right)$$

$$= - \left( \begin{matrix} \|w^*\| \\ \|w\| \end{matrix} \right)^T \left( \begin{matrix} \sin(2\theta) + 2\pi - 2\theta & -(2\pi - \theta)\cos(\theta) - \sin(\theta) \\ -(2\pi - \theta)\cos(\theta) - \sin(\theta) & 2\pi \end{matrix} \right) \left( \begin{matrix} \|w^*\| \\ \|w\| \end{matrix} \right)$$

$$\quad - \left( \begin{matrix} \|w^*\| \\ \|w\| \end{matrix} \right)^T \left( \begin{matrix} 2\pi - 2\theta & -(2\pi - \theta)\cos(\theta) \\ -(2\pi - \theta)\cos(\theta) & 2\pi \end{matrix} \right) \left( \begin{matrix} \|w^*\| \\ \|w\| \end{matrix} \right)$$

$$=: - \left( \begin{matrix} \|w^*\| \\ \|w\| \end{matrix} \right)^T (M_1 + M_2) \left( \begin{matrix} \|w^*\| \\ \|w\| \end{matrix} \right).$$

For $\theta \in (0, \pi/2)$, both $M_1, M_2$ are positive definite, and hence, the conclusion follows. □

**Theorem A.2** (Theorem 3). *Let $g(x; w) = (\sigma(w^T x))^2$ be a neural network with a single ReLU$^2$ node, where $w, x \in \mathbb{R}^d$, and $\sigma(x) = \max(0, x)$. We define the population loss function in $H^2$ space*

*as*

$$\mathcal{I}(w) = \mathbb{E}_X \left( \frac{1}{2N} \sum_{j=1}^N (g(x_j; w) - g(x_j; w^*))^2 + \|\nabla_x g(x_j; w) - \nabla_x g(x_j; w^*)\|^2 \right.$$

$$\left. + \|\nabla_x^2 g(x_j; w) - \nabla_x^2 g(x_j; w^*)\|^2 \right),$$

$$=: \mathcal{I}_1(w) + \mathcal{I}_2(w) + \mathcal{I}_3(w),$$

*for a teacher parameter $w^*$ and consider the gradient flow $\dot{w} = -\nabla_w \mathcal{I}(w)$. If $w^0 \in \{w : \|w - w^*\| < \|w^*\|\}$. Then,*

$$-(w - w^*)^T \nabla_w \mathcal{I}_j(w) < 0, \text{ for } j = 1, 2, 3,$$

*and hence, the convergence of $V = \|w - w^*\|^2$ is accelerated under the gradient flow that minimizes higher order Sobolev loss functions.*

*Proof.* We sequentially compute the analytical formulas of $\nabla_w \mathcal{I}_j(w)$.

$$\nabla_w \mathcal{I}_1(w) = \nabla_w \mathbb{E}\left( \frac{1}{2N} \sum_{j=1}^N (\sigma(w^T x_j)^2 - \sigma(w^{*T} x_j)^2)^2 \right)$$

$$= \mathbb{E}\left( \frac{1}{N} \sum_{j=1}^N (\sigma(w^T x_j)^2 - \sigma(w^{*T} x_j)^2) \nabla_w(\sigma(w^T x_j)^2) \right)$$

$$= \mathbb{E}\left( \frac{2}{N} \sum_{j=1}^N (\mathbb{1}_{w^T x_j > 0}(w^T x_j)^2 - \mathbb{1}_{w^{*T} x_j > 0}(w^{*T} x_j)^2) \mathbb{1}_{w^T x_j > 0}(w^T x_j) x_j \right)$$

$$= \mathbb{E}\left( \frac{2}{N} \sum_{w^T x_j > 0} (w^T x_j)^2 (w^T x_j) x_j - \sum_{\substack{w^T x_j > 0 \\ w^{*T} x_j > 0}} (w^{*T} x_j)^2 (w^T x_j) x_j \right)$$

Let $F(v, w) = \sum_{j=1}^N \mathbb{1}_{v^T x_j > 0 \wedge w^T x_j > 0}(v^T x_j)(w^T x_j)^2 x_j$, then $\nabla_w \mathcal{I}_1(w) = \frac{2}{N} \mathbb{E}(F(w, w) - F(w, w^*))$.

We consider an orthonormal basis $e = \frac{v}{\|v\|}$, $e_\perp = \frac{w/\|w\| - e \cos \theta}{\sin \theta}$, where $\theta = \angle(v, w)$, and any orthonormal set of vectors that span the rest. In this coordinate system, $e = (1, 0, \cdots 0), v = \|v\|e, w = (\|w\| \cos \theta, \|w\| \sin \theta, 0, \cdots, 0)$, and any vector $x = (r \cos \phi, r \sin \phi, z_3, \cdots z_d)$, where $\phi = \angle(x, e), r = \|x\|$. Then,

$\mathbb{E}(F(v, w))$

$$= N \int_{\mathbb{R}^{d-2}} \int_{-\frac{\pi}{2}+\theta}^{\frac{\pi}{2}} \int_0^\infty \|v\| r \cos \phi \|w\|^2 r^2 \cos^2(\phi - \theta) \begin{pmatrix} r \cos \phi \\ r \sin \phi \\ z_3 \\ \vdots \\ z_d \end{pmatrix} \frac{e^{-r^2/2}}{2\pi} r \, dr \, d\phi \, dz_3 \cdots dz_d$$

$$= \frac{N\|v\|\|w\|^2}{2\pi} \begin{pmatrix} \cos \theta(2 \sin \theta + 2(\pi - \theta) \cos \theta) + (\pi - \theta) + \sin \theta \cos \theta \\ \sin \theta(2 \sin \theta + 2(\pi - \theta) \cos \theta) \\ 0 \\ \vdots \\ 0 \end{pmatrix}$$

$$= \frac{N\|w\|^2}{2\pi}((\pi - \theta) + \sin \theta \cos \theta)v + \frac{N\|v\|\|w\|}{2\pi}(2 \sin \theta + 2(\pi - \theta) \cos \theta)w$$

Thus, $\nabla_w \mathcal{I}_1(w) = \left( 3\|w\|^2 w - \frac{\|w^*\|}{\pi}((\pi - \theta) + \sin \theta \cos \theta)w - \frac{\|w\|\|w^*\|}{\pi}(2 \sin \theta + 2(\pi - \theta) \cos \theta)w^* \right)$.
Now the first inequality follows. Let $G = (\pi - \theta) + \sin \theta \cos \theta$, $H = 2 \sin \theta + 2(\pi - \theta) \cos \theta$, then

the first result follows :

$$-(w - w^*)^T \nabla_w \mathcal{I}_1(w)$$

$$= -\frac{1}{\pi}(3\pi(\|w\|^2 - \|w\|\|w^*\|)^2 + 2\|w\|\|w^*\|(3\pi\|w\|^2 + 2\|w^*\|^2(G\cos\theta + H)$$

$$- 2\|w\|\|w^*\|(3\pi + G + H\cos\theta)))$$

$$= -\frac{1}{\pi}(3\pi(\|w\|^2 - \|w\|\|w^*\|)^2) - \frac{1}{2\pi}\begin{pmatrix}\|w^*\| \\ \|w\|\end{pmatrix}^T M \begin{pmatrix}\|w^*\| \\ \|w\|\end{pmatrix} < 0,$$

as

$$M = \begin{pmatrix} 2G\cos\theta + 2H & -(3\pi + G + H\cos\theta) \\ -(3\pi + G + H\cos\theta) & 6\pi \end{pmatrix}$$

is positive definite ($M_{11} > 0, M_{22} > 0, det(M) > 0$) for $\theta \in (0, \pi/2]$.

Now, we consider $\nabla_w \mathcal{I}_2(w)$. Note that $\nabla_x g(x; w) = 2\mathbb{1}_{w^T x > 0}(w^x)w$.

$$\nabla_w \mathcal{I}_2(w) = \nabla_w \mathbb{E}\left(\frac{1}{2N}\sum_{j=1}^N \|2\mathbb{1}_{w^T x_j > 0}(w^T x_j)w - 2\mathbb{1}_{w^{*T} x_j > 0}(w^{*T} x_j)w^*\|^2\right)$$

$$= \mathbb{E}\left(\frac{4}{N}\sum_{j=1}^N (\mathbb{1}_{w^T x_j > 0}(w^T x_j)(w^T w)x_j + \mathbb{1}_{w^T x_j > 0}(w^T x_j)^2 w\right.$$

$$- \mathbb{1}_{w^T x_j > 0 \wedge w^{*T} x_j > 0}(w^{*T} x_j)(w^T w^*)x_j$$

$$\left. - \mathbb{1}_{w^T x_j > 0 \wedge w^{*T} x_j > 0}(w^T x_j)(w^{*T} x_j)w^*)\right)$$

Let $F(v, w) = \sum_{j=1}^N \mathbb{1}_{v^T x_j > 0 \wedge w^T x_j > 0}(w^T x_j)((v^T w)x_j + (v^T x_j)w)$, then $\nabla_w \mathcal{I}_2(w) = \frac{4}{N}\mathbb{E}(F(w, w) - F(w, w^*))$. We again consider the orthonormal basis containing $e = \frac{v}{\|v\|}, e_\perp = \frac{w/\|w\| - e\cos\theta}{\sin\theta}$. Then,

$$\mathbb{E}(F(v, w))$$

$$= N\int_{\mathbb{R}^{d-2}}\int_{-\frac{\pi}{2}+\theta}^{\frac{\pi}{2}}\int_0^\infty \|v\|\|w\|r\cos(\phi-\theta)(\|w\|\cos\theta x_j + r\cos\phi w)\frac{e^{-r^2/2}}{2\pi}rdrd\phi dz_3\cdots dz_d$$

$$= \frac{N\cos\theta\sin\theta}{2\pi}\|w\|^2 v + \frac{N\|v\|\|w\|}{2\pi}(\sin\theta + 2(\pi-\theta)\cos\theta)w$$

Thus, $\nabla_w \mathcal{I}_2(w) = 4\left(\|w\|^2 w - \frac{\cos\theta\sin\theta}{2\pi}\|w^*\|^2 w - \frac{\|w\|\|w^*\|}{2\pi}(\sin\theta + 2(\pi-\theta)\cos\theta)w^*\right)$. Let $G_1 = \sin\theta + 2(\pi-\theta)\cos\theta$. Consequently,

$$-(w - w^*)^T \nabla_w \mathcal{I}_2(w)$$

$$= -\frac{2}{\pi}\left(2\pi\|w\|^4 - \cos\theta\sin\theta\|w^*\|^2\|w\|^2 - G_1\|w\|\|w^*\|\cos\theta\right.$$

$$\left. - 2\pi\|w\|^3\|w^*\|\cos\theta + \cos^2\theta\sin\theta\|w^*\|^3\|w\| + \|w\|\|w^*\|^3 G_1\right)$$

$$= -\frac{2}{\pi}\left(2\pi(\|w\|^2 - \|w\|\|w^*\|\cos\theta)^2 + \|w\|\|w^*\|\frac{1}{2}\begin{pmatrix}\|w^*\| \\ \|w\|\end{pmatrix}^T M_2 \begin{pmatrix}\|w^*\| \\ \|w\|\end{pmatrix}\right) < 0,$$

where

$$M_2 = \begin{pmatrix} 2G_1 + 2\cos^2\theta\sin\theta & -\cos\theta(G_1 + \sin\theta + 2\pi\cos\theta) \\ -\cos\theta(G_1 + \sin\theta + 2\pi\cos\theta) & 4\pi\cos\theta \end{pmatrix}$$

is positive definite for $\theta \in (0, \pi/2]$.

Finally, we prove $-(w - w^*)^T \nabla_w \mathcal{I}_3(w) < 0$. Note that $\nabla_x^2 g(x; w) = 2 \mathbb{1}_{w^T x > 0} w w^T \in \mathbb{R}^{d \times d}$.

$\nabla_w \mathcal{I}_3(w)$

$$= \nabla_w \mathbb{E}\left( \frac{1}{2N} \sum_{j=1}^N \| 2 \mathbb{1}_{w^T x_j > 0} w w^T - 2 \mathbb{1}_{w^{*T} x_j > 0} w^* w^{*T} \|^2 \right)$$

$$= \nabla_w \mathbb{E}\left( \frac{1}{2N} \sum_{j=1}^N trace\left( (2 \mathbb{1}_{w^T x_j > 0} w w^T - 2 \mathbb{1}_{w^{*T} x_j > 0} w^* w^{*T})^T (2 \mathbb{1}_{w^T x_j > 0} w w^T - 2 \mathbb{1}_{w^{*T} x_j > 0} w^* w^{*T}) \right) \right)$$

$$= \nabla_w \mathbb{E}\left( \frac{2}{N} \sum_{j=1}^N \left( \mathbb{1}_{w^T x_j > 0} \|w\|^4 - 2 \mathbb{1}_{w^T x_j > 0 \wedge w^{*T} x_j > 0} (w^T w^*)^2 + \mathbb{1}_{w^{*T} x_j > 0} \|w^*\|^4 \right) \right)$$

$$= \mathbb{E}\left( \frac{8}{N} \sum_{j=1}^N \left( \mathbb{1}_{w^T x_j > 0} \|w\|^2 w - \mathbb{1}_{w^T x_j > 0 \wedge w^{*T} x_j > 0} (w^T w^*) w^* \right) \right)$$

$$= \frac{8}{N} \sum_{j=1}^N \left( \mathbb{P}(w^T x_j > 0) \|w\|^2 w - \mathbb{P}(w^T x_j > 0 \wedge w^{*T} x_j > 0)(w^T w^*) w^* \right)$$

$$= 4\|w\|^2 w - \frac{4(\pi - \theta)}{\pi}(w^T w^*) w^*,$$

where $\theta$ denotes the angle between $w$, and $w^*$. Hence,

$$-(w - w^*)^T \nabla_w \mathcal{I}_3(w) = -\frac{4}{\pi}\left( \pi \|w\|^4 - (\pi - \theta)\|w\|^2 \|w^*\|^2 \cos^2 \theta - \pi \|w\|^3 \|w^*\| \cos \theta \right.$$
$$\left. + (\pi - \theta)\|w\|\|w^*\|^3 \cos \theta \right)$$
$$= -\frac{4}{\pi}\left( \pi(\|w\|^2 - w^T w^*)^2 + (w^T w^*) \left( \frac{\|w^*\|}{\|w\|} \right)^T M \left( \frac{\|w^*\|}{\|w\|} \right) \right) < 0,$$

where

$$M = \begin{pmatrix} 2(\pi - \theta) & (\theta - 2\pi)\cos\theta \\ (\theta - 2\pi)\cos\theta & 2\pi \end{pmatrix}$$

is positive definite and $w^T w^* > 0$ for $\theta \in (0, \frac{\pi}{2}]$. This completes the proof. $\qquad \square$

