# OpenReview forum: "Sobolev acceleration for neural networks"
_ICLR.cc/2024/Conference — Submitted to ICLR 2024_

### Official Review · Reviewer_BYTD · 2023-10-27

**Soundness:** 2 fair
**Presentation:** 3 good
**Contribution:** 2 fair
**Rating:** 6
**Confidence:** 3

**Summary:**

The authors study the effect of Sobolev acceleration for neural networks, focusing on the case of a ReLU network in the teacher student setting. They theoretically show that replacing the L2 loss by the corresponding Sobolev (H1 or H2) loss can increase of the rate of convergence. They consider the use of Chebyshev spectral differentiation to approximate the derivatives in the case where one does not have access to an analytic form. They also provide experiments which show that in a variety of settings, training using the Sobolev norm leads to faster and better convergence of the MSE loss.

**Strengths:**

- The experimental evidence for the speed up of Sobolev training seems quite strong
- In the experiments they also consider a setting beyond the teacher student setup, where the goal is to learn a general function the derivative of which is then approximated using Chebyshev spectral differentiation.
- The high-level presentation of their work is clear

**Weaknesses:**

The theoretical analysis seems a bit shallow, see also questions

**Questions:**

- In both theorem 2 and 3 it is written that  the convergence $\omega \to \omega^*$ is accelerated. How is this statement formally defined?
- Related to the above question, the bound on the derivative of $V$ in theorem 2 and 3 only shows that Sobolev training is at least as good as L2 training, but it is not clear how to derive any concrete quantitative bound from this result.
- In the beginning of section 4 it is stated that $\omega^*$ and $e$ are sampled randomly, what distributions are they sampled from?

---

> ### Author Response · Authors · 2023-11-13
> **Response to reviewer BYTD.**
>
> We appreciate your valuable comments.
>
> We will address your questions one by one:
>
> 1. In Theorem 2 and 3, $w\to w^\star$ is defined by $V(w) = \Vert w-w^\star \Vert_2^2 \to 0$ as $w$ follows the gradient flow $\dot{w} = -\nabla_w \mathcal{H}$. We apologize for the possible misleading.
>
> 2. Regarding Theorem 2, there was a typo in our original statement. The correct version is:
> $\frac{dV}{dt} = -(w-w^*)^T \nabla_w \mathcal{H} < -(w-w^*)^T \nabla_w \mathcal{L} <0,$ demonstrating that Sobolev training ensures strictly faster convergence.
>
> 3. For the sampling process, we generated $w^\star$ from the standard normal distribution and $e$ from the uniform distribution. We have updated our manuscript accordingly (refer to Section 4.1).
>
> We are open to further discussion. Please leave comments if you have any. Thank you.

---

### Official Review · Reviewer_ePTL · 2023-11-01

**Soundness:** 3 good
**Presentation:** 3 good
**Contribution:** 2 fair
**Rating:** 6
**Confidence:** 2

**Summary:**

This study investigates Sobolev training for neural networks, demonstrating its ability to expedite convergence and reduce test errors in comparison to traditional training methods, particularly for rectified linear unit (ReLU) networks within the student–teacher framework. By leveraging the analytical formula for population gradients of ReLU networks and extending findings through numerical examples to various network architectures, the paper enhances the understanding of ReLU network dynamics. Additionally, it introduces Chebyshev spectral differentiation as a solution for approximating target derivatives, addressing previous challenges and underscoring the convergence benefits, or "Sobolev acceleration," of this training approach.

**Strengths:**

Sobolev training is an interesting topic in recent years. Authors theoretically and empirically  showed the acceleration effect for a specific NN.
It's interesting to see authors proposed to approximate target function derivatives by using Chebyshev spectral differentiation

**Weaknesses:**

The works is restricted to a relatively simple architecture.
I am not sure its practical usage for other communities.

**Questions:**

See above

---

> ### Author Response · Authors · 2023-11-13
> **Response to reviewer ePTL.**
>
> We appreciate your kind consideration of our manuscript.
>
> We have revised our manuscript to address the weakness that the original manuscript is restricted to a relatively simple architecture.
> We have revised our manuscript to include the experimental results of the denoising autoencoder task with CNN applied to the MNIST dataset. Please refer to the revised manuscript and the official comment we provided.
>
> We are open to further discussion. Please leave comments if you have any.
> Thank you.

---

### Official Review · Reviewer_cgse · 2023-11-01

**Soundness:** 2 fair
**Presentation:** 2 fair
**Contribution:** 2 fair
**Rating:** 3
**Confidence:** 3

**Summary:**

The paper studies empirically, the effects of sobolev acceleration on training neural networks. It has some theoretical guarantees on very specific usecases, and has some experimental evidence.

**Strengths:**

The paper is quite interesting, and it has promise in training neural networks. It also studies various different non-linearities.

**Weaknesses:**

The paper studies, both theoretically and empirically, very niche use-cases, which do not represent real architectures or real datasets. This dramatically decreases the effect of the work, as it is not clear these results mirror the true reality of what goes on in deep neural networks.

**Questions:**

Could the authors provide results on more "real" dataset, e.g. CIFAR10, CIFAR100, Imagenet? Using modern architectures such as ConvNext, Resnet, RegNet, ViT? this will make the paper much stronger.

---

> ### Author Response · Authors · 2023-11-13
> **Response to reviewer cgse.**
>
> We appreciate your valuable comments.
>
> We agree that our original manuscript lacks experimental results demonstrating practical usage.
> Thus, we have revised our manuscript to include the experimental results of the denoising autoencoder task with CNN applied to the MNIST dataset.
> Please refer to the revised manuscript and the official comment we provided.
>
> We are open to further discussion. Please leave comments if you have any.
> Thank you.

---

### Official Review · Reviewer_dcHW · 2023-11-06

**Soundness:** 3 good
**Presentation:** 3 good
**Contribution:** 2 fair
**Rating:** 3
**Confidence:** 3

**Summary:**

This article presents a theoretical study of Sobolev Training, an alternative to least-squares fitting that replaces an $\mathcal{L}^2$ fitting error with $\mathcal{H}^p$ fitting error, for some $p > 0$. This method fits the target function by not only matching its values on the training data, but also matching higher order derivatives, leading to improved convergence speed.

To understand this effect, the authors study a toy model for training neural networks. They focus on one layer, single node RELU and RELU^2 networks, and they study the dynamics of model parameters under gradient flow-based minimization of the Sobolev loss. The Jacobian of the Sobolev loss has additional terms which are positive semidefinite when $w_0$ is sufficiently close to $w^*$, leading to (potentially) improved convergence speeds. These additional terms are not surprising, since the Sobolev loss can be thought of as a sum of $\mathcal{L}^2$ loss and additional terms measuring the $\mathcal{L}^2$ mismatch of gradients, which improve the convergence speed of gradient flow.

Finally, to approximate Sobolev Training in practice, the authors propose to use Chebyshev spectral differentiation, a popular numerical differentiation method. Across a variety of toy experiments, Sobolev Training is shown to accelerate convergence of GD and SGD.

**Strengths:**

- Simple idea and analysis: the idea behind Sobolev training is clearly presented and easy to understand. The authors identify a simple and straightforward theoretical explanation for empirical observations that Sobolev training improves convergence speed.
- Clear and correct proofs: the proofs in this work are presented clearly and they are correct to the best of my knowledge.

**Weaknesses:**

- Unfair/vacuous comparison: it is unfair to compare the convergence of gradient flow on $\mathcal{H}(w)$ and on $\mathcal{L}(w)$, since $\mathcal{H}(w)$ has extra terms that are added in an unnormalized way. To understand why this is unfair, consider another (trivial) way to accelerate the convergence speed of gradient flow, by rescaling the loss.  Set $\tilde{\mathcal{L}}(w) = \mathcal{L}(w) + \mathcal{L}(w)$, then in the notation of Theorem 1, $\partial \tilde{V} / \partial t = -2 (w - w^*)^T \nabla_w \mathcal{L} \leq 2 \partial V / \partial t < 0$, which is already a significantly stronger 'acceleration' than Theorems 2 and 3 because it converges twice as fast (whereas Theorem 2 and 3 show only convergence that is not any slower than Theorem 1). Simply adding extra convex terms to the loss isn't sufficient to argue an acceleration phenomena, any more than rescaling the loss would be. It would be more fair to average the terms in the Sobolev loss, or to rescale so that $\mathcal{L}(0) = \mathcal{H}(0)$, but then Theorems 2 and 3 might not hold.
- Synthetic/toy experiments: the experimental evaluation in this work is entirely focused training toy objectives. While the Sobolev loss is shown to improve convergence on these objectives, it's not clear whether this benefit extends to actual datasets.

**Questions:**

In Figure 3, Sobolev training seems to induce training instability in the form of large loss spikes during late training. Does the Sobolev training loss have higher variance over random initializations of GD? It would be helpful to add error bars to Figures 3, 4, and 5.

---

> ### Author Response · Authors · 2023-11-13
> **Response to reviewer dcHW.**
>
> We appreciate your time dedicated to reviewing our work and valuable comments.
>
> We acknowledge that one may perceive the theoretical results as suggesting an unfair comparison. However, our primary objective in presenting the theorems is to highlight the convergence acceleration of Sobolev training in the $H^1$-norm compared to $L^2$-training. It's worth noting that the specific example $\tilde{\mathcal{L}}(w) = 2\mathcal{L}(w)$  you mentioned may seem to accelerate convergence, but this effect arises from the infinitesimal learning rate assumption and the student-teacher assumption in theoretical settings. Moreover, comparing $L^2$ and $H^1$ loss functions is a standard practice widely used in the literature [1,2].
>
> In summary, our focus is on demonstrating the acceleration of $H^1$-training rather than suggesting a new loss function (such as $\tilde{\mathcal{L}}(w)$) for faster convergence. We hope you understand our intent.
>
> In regards to experiments, we have revised our manuscript to include more practical experiments. Please refer to the updated manuscript and the official comment we have provided.
>
> We are open to further discussion. Please leave comments if you have any.
> Thank you.
>
>
> [1] Wojciech M Czarnecki, Simon Osindero, Max Jaderberg, Grzegorz Swirszcz, and Razvan Pascanu. Sobolev training for neural networks. Advances in neural information processing systems, 30, 2017.
>
> [2] Hwijae Son, Jin Woo Jang, Woo Jin Han, and Hyung Ju Hwang. Sobolev training for physics-informed neural networks. Communications in Mathematical Sciences, 2023.

---

> > ### Comment · Reviewer_dcHW · 2023-11-16
> > **Response to rebuttal**
> >
> > Thank you for taking the time to answer my questions and to add experiments. I appreciate the clarification in your intent behind writing the paper. I feel that my score remains an accurate assessment of this work and I am not currently planning to change it.

---

### Author Response · Authors · 2023-11-13
**Revised manuscript is now available**

Dear Reviewers,

We sincerely appreciate the time and thoughtful feedback provided on our manuscript. The valuable comments have been instrumental in enhancing the quality of our work.

In response to the common concern raised by the reviewers regarding the original manuscript's focus on synthetic experiments, we have incorporated a practical example: the denoising autoencoder task with Convolutional Neural Networks(CNN). This example is discussed in [1] along with Sobolev training. The experiments demonstrate both the convergence acceleration and improved generalization ability of Sobolev training.

We are in the process of preparing detailed responses to each reviewer's comments. Thank you once again for your constructive input.

Best regards.

[1] Annan Yu, Yunan Yang, and Alex Townsend. Tuning frequency bias in neural network training with nonuniform data. In The Eleventh International Conference on Learning Representations, 2023.

---

### Meta-Review · Area_Chair_cKfo · 2023-12-05

**Metareview:**

This paper shows theoretically that learning, Sobolev training, accelerates learning of ReLU activation, and also shows numerically that it occurs in complex networks. The theoretical analysis uses the sign of error dynamics. The paper is clearly written and accurately conveys the authors' intentions. However, the theoretical analysis method and results are judged to be reasonably natural and not surprising enough to accept. Further work on the theory, such as considering a multi-layered structure (with some simplifications), other classes of activation functions, or better quantifying the speed of convergence, would improve the value of this paper.

**Justification For Why Not Higher Score:**

The weaknesses of this paper are clearly identified and there is consensus among the reviewers that there is room for improvement.

**Justification For Why Not Lower Score:**

N/A

---

### Decision · Program_Chairs · 2024-01-16

Reject